# Ballroom Dance Movement Recognition Using a Smart Watch and Representation Learning

## Abstract

Smart watches are being increasingly used to detect human gestures and movements. Using a single smart watch, whole body movement recognition remains a hard problem because movements may not be adequately captured by the sensors in the watch. In this paper, we present a whole body movement detection study using a single smart watch in the context of ballroom dancing. Deep learning representations are used to classify well-defined sequences of movements, called *figures*. Those representations are found to outperform ensembles of decision trees and hidden Markov models. The classification accuracy of 85.95% was improved to 92.31% by modeling a dance as a first-order Markov chain of figures.

## 1 Introduction

Recent work has used low-cost smart watches to track the movement of human body parts. ArmTrak tracks arm movement, assuming that the body and torso are stationary (Shen et al., 2016). In this paper, we perform whole body movement recognition using a single smart watch, which is a hard problem given that body movements need to be inferred using readings taken from a single location on the body (the wrist). The movements in the study are from ballroom dancing, which engages tens of thousands of competitors in the U.S. and other countries. Competitors dance at different skill levels and each level is associated with an internationally recognized syllabus, set by the World Dance Sport Federation. The syllabus breaks each dance into smaller segments with well-defined body movements. Those segments are called *figures*. In the waltz, for example, each figure has a length of one measure of the waltz song being danced to; the entire dance is a sequence of 40 to 60 figures (depending on the length of the song). The sequence is random, but the figures themselves are well-defined. The sequence is illustrated in Fig. 1.

The International Standard ballroom dances are a subset of ballroom dances danced around the world, and they include the waltz, tango, foxtrot, quickstep and Viennese waltz. A unique characteristic of all these dances is that the couple is always in a closed-hold, meaning they never separate. Also, both dancers in the couple maintain a rigid frame, meaning the arms and torso move together as one unit. The head and the lower body, however, move independently of that arms-torso unit. Our hypothesis in this paper is that the figures in each of these dances can be recognized with high accuracy using deep learning representations of data obtained from a single smart watch worn by the lead in the couple. That is possible because the rigid frame makes it unnecessary to separately instrument the arms and torso, and because most figures are characterized by distinct movements (translations and rotations in space) of the arms and torso. We refer the interested reader to the website `www.ballroomguide.com` for free videos and details on the various syllabus figures in all the International Standard ballroom dance styles.

In this paper, we validate our hypothesis on the quintessential ballroom dance– the waltz. We chose 16 waltz figures that are most commonly danced by amateurs. The full names of the figures are included in Appendix A. Our goal is to accurately classify those figures in real-time using data from a smart watch. That data can be pushed to mobile devices in the hands of spectators at ballroom competitions, providing them with real-time commentary on the moves that they will have just watched being performed. That is an augmented-reality platform serving laymen in the audience who want to become more engaged with the nuances of the dance that they are watching.

The main beneficiary of the analysis of dance movements would be the dancers themselves. The analysis will help them identify whether or not they are dancing the figures correctly. If a figure

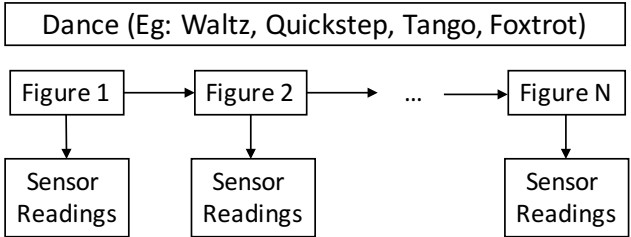

Figure 1: A dance is random a sequence of well-defined figures (movements). If the dancer is instrumented with sensors, the figures emit sensor readings that should be similar for each type of figure.

is confused for a different figure, it may be because the dancers have not sufficiently emphasized the difference in their dancing and need to improve their technique on that figure. That confusion metrics could also be used by competition judges to mark competitors on how well dancers are performing figures; that task is currently done by eye-balling multiple competitors on the floor, and is challenging when there are over ten couples to keep track of.

We make three main contributions in detecting ballroom dance movements using learning representations.

- First, we show that representations using data from a single smart watch are sufficient for discriminating between complex dancing movements.

- Second, we identify and evaluate six learning representations that can be used for classifying the figures with varying accuracies. The representations are 1) Gaussian Hidden Markov Model, 2) Extra Trees Classifier, 3) Feed-Forward Neural Network, 4) Recurrent Neural Network (LSTM), 5) Convolution Neural Network, and 6) a Convolution Neural Network that feeds into a Recurrent Neural Network.

- Finally, we model the sequence of figures as a Markov chain, using the fact that the transitions between figures are memoryless. We use the rules of the waltz to determine which transitions are possible and which are not. With that transition knowledge, we correct the immediately previous figure's estimate. This leads to an average estimation accuracy improvement of 5.33 percentage points.

## 2 DATASET DESCRIPTION

### 2.1 DATA COLLECTION

The data was collected using an Android app on a Samsung Gear Live smart watch. The app was developed for this work on top of the ArmTrak data collection app. We were able to reliably collect two derived sensor measurements from the Android API:

- Linear Acceleration. This contains accelerometer data in the X, Y and Z directions of the smart watch, with the effect of gravity removed.

- Rotation Vector. This provides the Euler angles (roll, pitch and yaw) by fusing accelerometer, gyroscope and magnetometer readings in the global coordinate space. We use only the yaw (rotation about the vertical axis) in this study, and that is based on prior knowledge that roll and pitch are insignificant in the waltz figures included in the study.

In total, we collected readings from 4 sensor axes (three from the Linear Acceleration and the yaw from the Rotation Vector sensors). The readings were reported by watch operating system asynchronously, at irregular intervals, whenever a change was sensed. In order to facilitate signal processing, we downsampled the data such that each figure contained exactly 100 sensor samples, which was possible because the effective sampling rate was greater than that. The downsampling was done by taking the median (instead of the mean, which is sensitive to outliers) values of 100

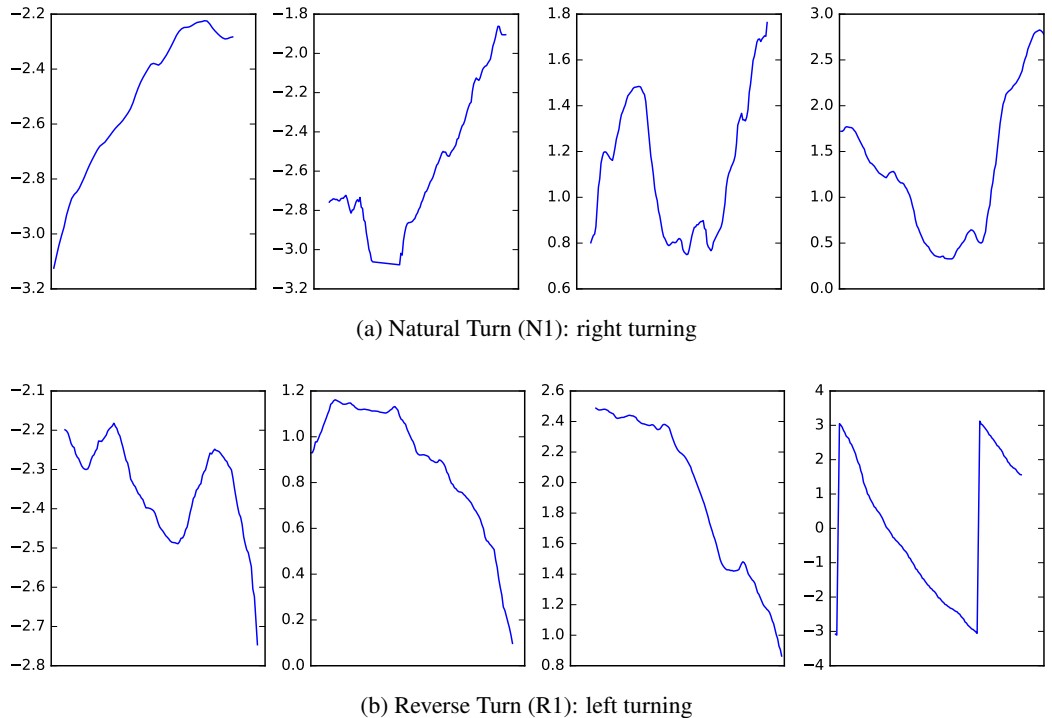

(a) Natural Turn (N1): right turning

(b) Reverse Turn (R1): left turning

Figure 2: Raw measurements of yaw: 4 samples from two figures.

evenly-spaced time windows. From this point on in the paper, when we refer to "samples", we refer to an observation for a figure of dimension $4 \times 100$ as one sample.

The app was developed in such a way that the button on the watch that started recording the afore-mentioned sensor measurements also simultaneously started playing the music via Bluetooth speakers. That ensured that the music and the recording of the movements were time-synchronized.

For all the data that was collected, we used the same rendition of the classic song "Moon River". We performed manual segmentation of the song using its beats offline, and that was used to segment the time series data for the entire dance sequence into 2.1-second-long-segments corresponding to figures in the dance. We noted the song intro length (where no dancing was performed) and ignored all data in that period. For each figure, we extended the window of measurements equally at the beginning and at the end by 0.35 seconds to account for slight errors in dancer timing. That ensured that the window captures the figures even if the dancer was slightly early or late to begin/finish dancing the figure.

The yaw data for 4 figure samples corresponding to two different figures are illustrated in Fig. 2. It can be seen that right-turning figures tended to record yaw readings with an upward trend, while left-turning figures recorded yaw readings with a downward trend. Slight differences between the samples for each figure can be attributed to differences in the dancers' timing and execution.

## 2.2    CROSS-VALIDATION GROUPINGS

In total, we collected 818 figure samples across 16 different waltz figures, over 14 dances (figure sequences). Thus, the input data had a dimension of $\Re^{818 \times 4 \times 100}$ for 818 figure samples, 4 sensors, and 100 measurements per sensor per figure sample.

The small size of the dataset and subsequent difficulty in collecting additional data during the Covid-19 pandemic made the learning problem more challenging. We had only 14 sequences in total, and that is too small of dataset to learn dependencies between figures. Therefore, our focus is on independently classifying the figures using the 818 samples, and leveraging the Markov property to enforce dependencies between figures.

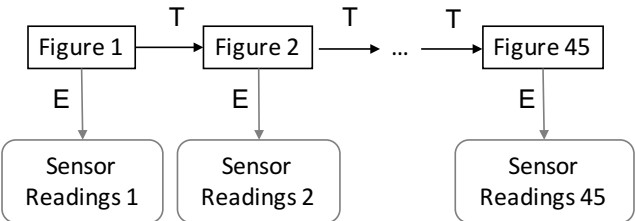

Figure 3: Hidden Markov Model representation.

The 818 samples came from 14 separate dances (figure sequences) and we performed 7-fold cross-validation with two dances per cross-validation group (assigned randomly). That ensured sequences of figures (dances) were not split across different cross-validation groups. It also allowed us to test our representations' accuracy for each sequence as a whole.

## 2.3 LABELING GROUND TRUTH

Each dance was recorded on video so that labels (ground truth) could be given to the data segments corresponding to the figures. The labels are listed in Appendix A.

## 3 MARKOV TRANSITIONS

The sequence of figures in each dance can be modeled as a Markov chain. The probability of observing the next figure is dependent on the current figure, but independent of past figures given the current figure. The reason is as follows.

Certain figures end on the right foot, while others end on the left foot. Similarly, certain figures begin on the left foot, while others begin on the right foot. The probability of going from a figure ending on the right foot to another figure beginning on the right foot is zero (and the same applies to the left foot). That is because of the physics of the dance and the way weight is distributed between the feet. Similarly, some figures must be followed by figures that move forward while others must be followed by figures that go backward. Therefore, each figure constrains the immediate next figure, but the sequence is memoryless.

Using the above rules, we constructed a transition matrix for all figures, and that is given in the Appendix in Table 3. We essential gave a zero probability to impossible transitions, and equal probability to all possible transitions. Therefore, our transition matrix is completely unbiased, and not based on real training data. The advantage of the unbiased transition matrix is that the same matrix can be used across different couples since it is very general. It does not encode unique habits of certain couples, where there is a tendency to follow patterns. If a biased approach were taken, a unique transition matrix could be learned for each couple, but it would not generalize to other couples.

## 4 HIDDEN MARKOV MODEL REPRESENTATION

The dance can be represented as a Hidden Markov Model (HMM) where the states represent figures that emit sensor readings, as illustrated in Fig. 3. Although the state space is discrete, the emission space is continuous because the sensor readings continuous. As a result, the HMM cannot be solved using a discrete emission probability matrix. Instead, we assumed Gaussian emission probabilities, resulting in a Gaussian HMM.

We used the HMMLearn Python library (hmm) to estimate the transition and emission probabilities while fitting the input data. We initialized the transition probabilities with the trained transition matrix described in Section 3, and initialized state vector with the actual initial state obtained from the ground truth.

The problem with the HMM approach for this task is that the HMM is a generative model, and not a discriminative model. At no stage does the model take the actual known labels to perform

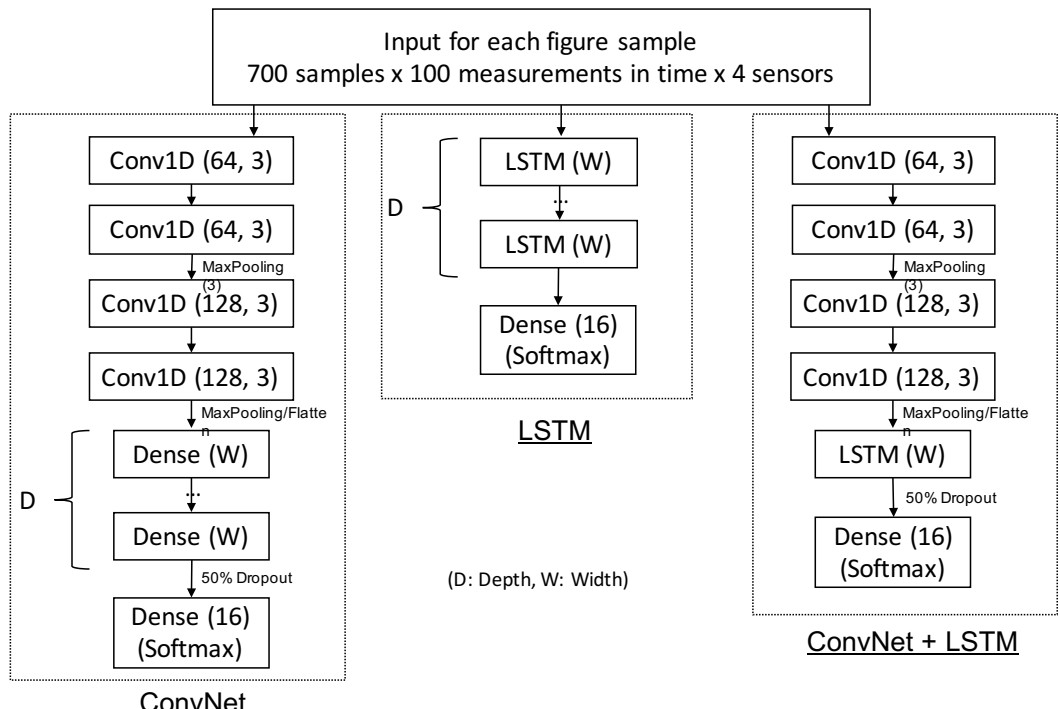

Figure 4: Deep Neural Networks Representations.

classification. It simple estimates states using the probability information and we assigned labels to the states by fitting the training set, and matching the states estimated by the HMM with the known labels. The approach achieved an accuracy of 35.93% on the validation sets, averaged across the 7 cross validation groups.

## 5 DECISION TREE REPRESENTATION

We used the Extra Trees Classifier provided in the Scikit-Learn Python library (Pedregosa et al., 2011) to classify figures directly from the downsampled data. That classifier is an ensemble method incorporating several (250 in our case) decision trees and aggregating their results. Each input sample was in $\Re^{400}$ (4 sensors, 100 time series points per sensor), so there were 400 features. The approach achieved an average accuracy of 72.2%.

## 6 DEEP NEURAL NETWORKS REPRESENTATIONS

We tested three different deep neural network architectures, illustrated in Fig. 4 and a standard feed-forward neural network (not illustrated). In all layers, we used ReLu activations, except for the LSTM layers for which we used Sigmoid activations. The inputs to all the networks are the same, and are based on the cross-validation groupings described in Section 2.2. The outputs are also the same, because we want to obtain the probabilities associated with the different figures. Therefore, we use a softmax output layer with a categorical cross-entropy loss function.

We used the Keras (Ker, 2017) package for Python, which provides an abstraction for a Tensor-flow (Abadi et al., 2015) backend. We trained the networks using the Adam solver (Kingma & Ba, 2014).

- Feed-forward: All layers are densely connected. There are D dense layers, each of width $W$. We varied $D$ and $W$, as given in Appendix C.
- Convolutional (ConvNet): Since we are looking at 4 1-dimensional streams, we used 1-dimensional convolution layers. The first two layers are convolutional with 64 filters and a

Table 1: Results for Mean Accuracy with 7-fold Cross-Validation (%)

| | Classifier Only | Classifier + Markov Correction |
|---|---|---|
| **Random Guess** | 6.25 | N.A. |
| **Gaussian HMM** | 35.93 | N.A. |
| **Extra Trees Classifier** | 72.20 | 73.48 |
| **Feed-forward** | 80.86 | 85.63 |
| **ConvNet** | 83.01 | 88.90 |
| **LSTM** | 83.73 | 92.31 |
| **ConvNet +LSTM** | 85.95 | 88.41 |

kernel size of 3. The next two layers are preceded by a max pooling operation, and contain 128 filters and a kernel size of 3. Another max pooling operation is added before D dense layers of width $W$. This particular architecture was inspired from the example in (Ker, 2017).

- Recurrent (LSTM): This layer has D Long Short-term Memory (LSTM) layers (Hochreiter & Schmidhuber, 1997). The layers have $W$ nodes with a time history of 100 and 4 features each (as per the input dimension). This network took the longest time to train because it had the most parameters.

- Hybrid Convolutional and Recurrent (ConvNet+LSTM): This network is a hybrid of the aforementioned convolutional and recurrent architectures. One LSTM layer replaces the dense layers in the convolutional architecture. The complexity of this layer is less than that of the pure LSTM network because the convolutional layers reduce the input dimensionality. As a result, this network trains faster than the pure LSTM architecture. This hybrid architecture was inspired from related work (Morales & Roggen, 2016).

As described previously, we did not attempt to learn the entire sequence because of the number of sequences was too small. Instead, we focused on representations for classifying each figure independently.

## 7 MARKOV CORRECTION

In this section, we propose a simple approach to combine the results of the learning representations (referred to as *classifiers*) with the Markov structure of the dance. Let $i, j \in 0, 1, ..., 15$ be possible state from the 16 different figures, and let $X_t$ be the figure at time index $t$. Then at each time index $t$,

1. We assume that the classifier is correct for the current figure and suppose $X_t = j$
2. We correct the immediately preceding figure $X_{t-1}$ as follows.

$$X_{t-1} = \arg \max_i P(X_{t-1} = i | X_t = j)$$
$$= \arg \max_i P(X_t = j | X_{t-1} = i) P(X_{t-1} = i)$$

We get $P(X_t = j | X_{t-1} = i)$ from the trained transition matrix described in Section 3 and $P(X_{t-1} = i)$ from the classifier.

## 8 EVALUATION RESULTS

The evaluation was performed with 7-fold cross-validation with groupings described in Section 2.2. The results are summarized below and include the best configurations for the neural networks. The results for all the different configurations are given in Appendix C. There is no directly related work that can be used for comparative evaluation. However, the accuracies presented can be compared with the accuracy of a random guess, which is $\frac{1}{16} = 6.25\%$.

From the results, it is clear that the neural networks approaches outperform the Extra Trees Classifier (ensemble of decision trees). The hybrid approach with the convolutional and LSTM layers performs

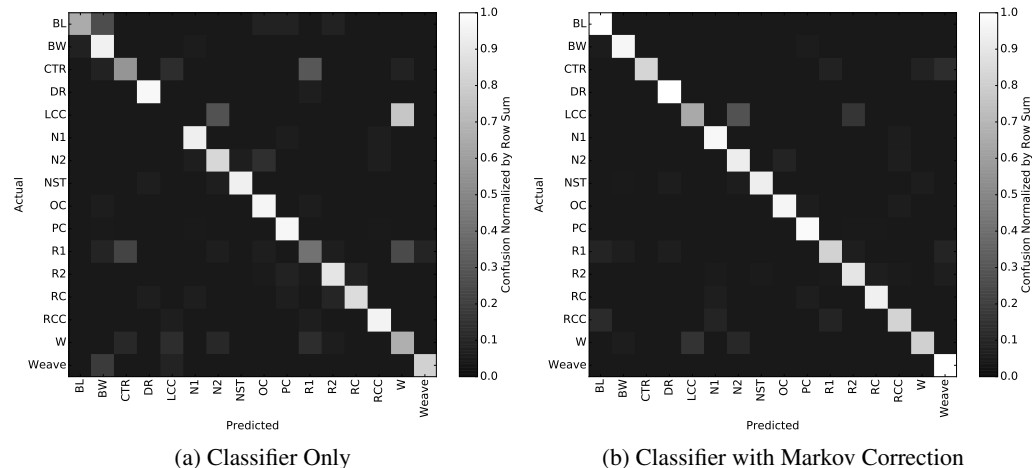

(a) Classifier Only  (b) Classifier with Markov Correction

Figure 5: Confusion matrices. When the actual figure is correctly classified p% of the time, the diagonal entry is p/100.

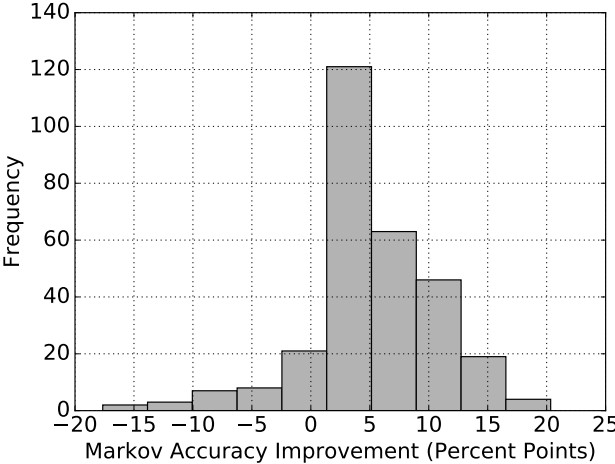

Figure 6: Improvement achieved using Markov correction across all neural network configurations and cross-validation test sets.

the best. It is also clear that the number of samples (figures) in the training sets was sufficient to achieve good performance on the validation sets. Collecting more data may have improved the performance further, but despite data collection constraints, the Markov correction provided the required boost in performance.

On average, the Markov correction approach proposed in Section 7 is found to benefit all the classifiers. We illustrate this using confusion matrices, which capture the results for individual figures. Ideally, the confusion matrix should be the $16 \times 16$ identity matrix, because that would mean that the predicted figure was always the actual figure. However it can be seen in Fig. 5(a) that the left-foot closed-change (LCC) is most often confused for a whisk (W). However, a whisk is almost always followed by a progressive chasse (PC). The Markov correction approach recognizes this from the transition probabilities and corrects the estimation of a whisk to a left-foot closed change as soon as it sees that that figure was not followed by a progressive chasse. The improved classification results is illustrated in Fig. 5(b).

Markov correction sometimes hurts the classification results because the assumption that the current figure was correct may not be valid. If the current figure has been incorrectly classified, then that error in classification could be propagated to the previous figure. Fig. 6 shows the distribution of improvements. On average, the improvement was 5.33 percentage points.

## 9 RELATED WORK

To the best of our knowledge, we are the first to use a smart watch for dance recognition and dance analytics. Multiple accelerometers are used as input for dancing video games in (Crampton et al., 2007). VICON systems were proposed in (Dyaberi et al., 2004) and video recognition was used in (Matthew Faircloth, 2008). Those approaches do not work in our scenario where there are multiple ballroom dancers simultaneously on the floor, leading to occlusion. Also, they are expensive and not suited for amateurs.

Music segmentation studies for dance detection purposes are presented in (Shiratori et al., 2004). Models for turning motions in Japanese folk dances are modeled from observation in (Rennhak et al., 2010). Signal processing techniques used in dance detection are reviewed in (Pohl, 2010). Ballroom dance styles are differentiated in (Schuller et al., 2008) from the music that is being played.

For human activity recognition, ensembles of deep LSTM networks were proposed in (Guan & Plötz, 2017), but this approach is not suitable for real-time prediction because it is too slow. A single deep LSTM network took nearly a whole day to train, from our experiments, and loading the weights for prediction was also very slow. Convolutional neural networks were proposed in (Zeng et al., 2014). Our approach is similar, but we use more convolutional layers, as suggested in the Keras time series classification example (Ker, 2017). Our best results were obtained using the hybrid architecture between convolutional and recurrent neural networks, and that was proposed for human activity recognition in (Morales & Roggen, 2016).

More generally, for time series classification, convolutional networks were proposed in (Zhao et al., 2017) and (Cui et al., 2016).

## 10 CONCLUSION

In this paper, we presented a study of whole body movement detection using a single smart watch in the context of competitive ballroom dancing. Our approach was able to successfully classify movement segments from the International Standard Waltz, using deep learning representations. The representations alone achieved a maximum accuracy of 85.95%, averaged over 7 cross-validation groups. Using the fact that the segments can be represented as a Markov chain, the accuracy was improved to 92.31% by correcting the prediction for each preceding segment. The deep learning representations outperformed ensembles of decision trees, and a Gaussian HMM representation performed poorly because it was not discriminative. Despite the small size of the training set, the representations were able to generalize well to validation sets.

ACKNOWLEDGMENTS

Redacted for peer review.

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

## A  APPENDIX: WALTZ FIGURE INFORMATION

Ballroom dancing competitions in the U.S. are conducted at multiple skill levels. The syllabus at the skill levels restricts which figures can be used. In this paper, we focus on Waltz figures at the Newcomer and Bronze skill levels, because the majority of amateur dancers compete at these skill levels. The figures can also be used by dancers at more advanced skill levels, such as Silver and Gold. Table 2 gives the names of the figures that we consider in this paper, along with the short names used throughout the paper.

Table 2: Waltz Figure Names

| Left Foot Figures | Right Foot Figures |
|---|---|
| Left-foot Closed Change (LCC) | Right-foot Closed Change (RCC) |
| Natural Turn 4-6 (N2) | Natural Turn 1-3 (N1) |
| Natural Spin Turn (NST) | Reverse Corte (RC) |
| Reverse Turn 1-3 (R1) | Reverse Turn 4-6 (R2) |
| Chasse to Right (CTR) | Chasse from Promenade (PC) |
| Outside Change (OC) | Basic Weave (Weave) |
| Double Reverse (DR) | |
| Whisk (W) | |
| Back Whisk (BW) | |
| Back Lock (BL) | |

## B  UNIFORM (UNBIASED) TRANSITION PROBABILITIES

Table 3 describes the uniform (unbiased) transition probabilities between figures of the international standard waltz.

Table 3: Figure Transition Probabilities

|       | BL   | BW   | CTR  | DR   | LCC  | N1   | N2   | NST  | OC   | PC   | R1   | R2   | RC   | RCC  | W    | Weave |
|-------|------|------|------|------|------|------|------|------|------|------|------|------|------|------|------|-------|
| BL    | 0.20 | 0.20 | 0.00 | 0.00 | 0.00 | 0.00 | 0.20 | 0.20 | 0.20 | 0.00 | 0.00 | 0.00 | 0.00 | 0.00 | 0.00 | 0.00  |
| BW    | 0.00 | 0.00 | 0.00 | 0.00 | 0.00 | 0.00 | 0.00 | 0.00 | 0.00 | 1.00 | 0.00 | 0.00 | 0.00 | 0.00 | 0.00 | 0.00  |
| CTR   | 0.20 | 0.20 | 0.00 | 0.00 | 0.00 | 0.00 | 0.20 | 0.20 | 0.20 | 0.00 | 0.00 | 0.00 | 0.00 | 0.00 | 0.00 | 0.00  |
| DR    | 0.00 | 0.00 | 0.20 | 0.20 | 0.20 | 0.00 | 0.00 | 0.00 | 0.00 | 0.00 | 0.20 | 0.00 | 0.00 | 0.00 | 0.20 | 0.00  |
| LCC   | 0.00 | 0.00 | 0.00 | 0.00 | 0.00 | 0.33 | 0.00 | 0.00 | 0.00 | 0.33 | 0.00 | 0.00 | 0.00 | 0.33 | 0.00 | 0.00  |
| N1    | 0.20 | 0.20 | 0.00 | 0.00 | 0.00 | 0.00 | 0.20 | 0.20 | 0.20 | 0.00 | 0.00 | 0.00 | 0.00 | 0.00 | 0.00 | 0.00  |
| N2    | 0.00 | 0.00 | 0.00 | 0.00 | 0.00 | 0.33 | 0.00 | 0.00 | 0.00 | 0.33 | 0.00 | 0.00 | 0.00 | 0.33 | 0.00 | 0.00  |
| NST   | 0.00 | 0.00 | 0.00 | 0.00 | 0.00 | 0.00 | 0.00 | 0.00 | 0.00 | 0.00 | 0.00 | 0.33 | 0.33 | 0.00 | 0.00 | 0.33  |
| OC    | 0.00 | 0.00 | 0.00 | 0.00 | 0.00 | 0.33 | 0.00 | 0.00 | 0.00 | 0.33 | 0.00 | 0.00 | 0.00 | 0.33 | 0.00 | 0.00  |
| PC    | 0.00 | 0.00 | 0.00 | 0.00 | 0.00 | 0.33 | 0.00 | 0.00 | 0.00 | 0.33 | 0.00 | 0.00 | 0.00 | 0.33 | 0.00 | 0.00  |
| R1    | 0.00 | 0.00 | 0.00 | 0.00 | 0.00 | 0.00 | 0.00 | 0.00 | 0.00 | 0.00 | 0.00 | 0.33 | 0.33 | 0.00 | 0.00 | 0.33  |
| R2    | 0.00 | 0.00 | 0.20 | 0.20 | 0.20 | 0.00 | 0.00 | 0.00 | 0.00 | 0.00 | 0.20 | 0.00 | 0.00 | 0.00 | 0.20 | 0.00  |
| RC    | 0.20 | 0.20 | 0.00 | 0.00 | 0.00 | 0.00 | 0.20 | 0.20 | 0.20 | 0.00 | 0.00 | 0.00 | 0.00 | 0.00 | 0.00 | 0.00  |
| RCC   | 0.00 | 0.00 | 0.20 | 0.20 | 0.20 | 0.00 | 0.00 | 0.00 | 0.00 | 0.00 | 0.20 | 0.00 | 0.00 | 0.00 | 0.20 | 0.00  |
| W     | 0.00 | 0.00 | 0.00 | 0.00 | 0.00 | 0.00 | 0.00 | 0.00 | 0.00 | 1.00 | 0.00 | 0.00 | 0.00 | 0.00 | 0.00 | 0.00  |
| Weave | 0.20 | 0.20 | 0.00 | 0.00 | 0.00 | 0.00 | 0.20 | 0.20 | 0.20 | 0.00 | 0.00 | 0.00 | 0.00 | 0.00 | 0.00 | 0.00  |

## C  APPENDIX: DETAILED RESULTS FOR NEURAL NETWORK HYPERPARAMETER CONFIGURATIONS

The following table contains the detailed results and the number of model parameters for each of the architectures described in Fig 4. Note that $D$ refers to the number of hidden layers for the feed-forward neural network, but it refers to the number of LSTM/Dense layers in the other architectures (described in Fig 4)

| Architecture | Width (W) | D | Classifier Accuracy (%) | Markov Correction (%) | Model Parameters |
|---|---|---|---|---|---|
| ConvNet | 500 | 1 | 81.07 | 87.58 | 671,684 |
|  | 500 | 2 | 80.19 | 85.46 | 922,184 |
|  | 1000 | 1 | 83.01 | 87.98 | 1,256,184 |
|  | 1000 | 2 | 82.30 | 87.39 | 2,257,184 |
|  | 2000 | 1 | 82.40 | 86.46 | 2,425,184 |
|  | **2000** | **2** | **82.04** | **88.90** | 6,427,184 |
| ConvNet+LSTM | 1000 | 1 | 84.28 | 89.51 | 4,619,184 |
|  | *2000* | *1* | *85.95* | *92.31* | *17,151,184* |
|  | 3000 | 1 | 85.93 | 91.86 | 37,683,184 |
| Feed-Forward | 500 | 1 | 78.87 | 85.27 | 208,516 |
|  | **500** | **2** | **80.14** | **85.63** | 459,016 |
|  | 500 | 3 | 79.04 | 83.10 | 709,516 |
|  | 1000 | 1 | 78.17 | 83.82 | 417,016 |
|  | 1000 | 2 | 78.73 | 84.23 | 1,418,016 |
|  | 1000 | 3 | 79.76 | 85.05 | 2,419,016 |
|  | 2000 | 1 | 78.77 | 83.80 | 834,016 |
|  | 2000 | 2 | 80.86 | 84.90 | 4,836,016 |
|  | 2000 | 3 | 76.83 | 80.98 | 8,838,016 |
| LSTM | 500 | 1 | 83.73 | 88.26 | 1,018,016 |
|  | **1000** | **1** | **83.19** | **88.41** | 4,036,016 |
|  | 2000 | 1 | 78.38 | 84.85 | 16,072,016 |

