# OpenReview forum: "Ballroom Dance Movement Recognition Using a Smart Watch and Representation Learning"
_ICLR.cc/2021/Conference — Reject_

### Official Review · AnonReviewer2 · 2020-10-28
**Interesting application but not enough novelty**

**Rating:** 4
**Confidence:** 3

**Review:**

The paper presents some classification results for ballroom dancing movements, as measured by inertial sensors on a smartwatch.  The motivation is mixed - as a guide to dancers themselves and as an automatic grading mechanism for competition judges. However the sensors used can only measure a very limited aspect of the dance, and there is no 'gold standard' data describing the full motion of the dancers that can then be compared with the limited data actually measured, to be able to contrast variations in the full body motion with variations in the intertial sensors on the smartwatch.  Describing the project as a study of whole body movement seems a bit brave given that you measure only hand movements. The restriction to the yaw axis will make results very sensitive to how the hand is held.

The paper is fairly clearly described, but the work uses fairly standard model structures and a very limited training set, so I'm not sure the reader really learns anything novel from the paper. In my opinion, it is below the level of technical challenge and novelty of solution usually associated with an ICLR paper

It is unclear whether cross-validation splitting was done randomly rather than breaking the time-series into a number of contiguous sections and working with these. If randomly from whole set, then I would worry about  effective independence. The training data seems to be very limited, with little variability in individuals, so it is hard to know whether the results are of any practical use.

---

### Official Review · AnonReviewer4 · 2020-10-28

**Rating:** 4
**Confidence:** 5

**Review:**

Authors propose an approach to perform classification of ballroom dance movements (called figures) captured by the sensing mechanism of a smartwatch and discriminated via different ANN architectures. The sequence of figures are modelled as a Marlov chain, which work in  a generative+discriminative fashion to output the final prediction of the model. Authors also present a dataset collected specifically for this work, to perform the inference of the algorithms included in the evaluation. Results show a remarkable accuracy, but are not compared to any existing state of art due to limited related work.

-------------------------------------------------
Contributions

The paper addresses an application of HAR which I see can be of interest for the community due to its novelty. The dance recognition it’s not a common domain and its analysis in terms of feature representation can fit correctly within the scope of the conference.

The paper is well written and easy to read. The state of art is adequate and gives enough background on the domain

-------------------------------------------------
Points to improve

The principal flaw of the work, and main reason why I do not recommend its acceptance in the proceddings is its weak experimental setup and overall evaluation. Other, but less important, reasons for my recommendation are the lack of comprehensive reproducibility details and the weak findings included in the conclusions.

-------------------------------------------------
Recommendations

A single dance (one type of sequence), performed by a single couple (lack of variability in the data collected) and a predefined sensor system (lack of multimodal/multilocation setup) is not sufficient to achieve solid results for the type of algorithm the authors want to evaluate. I understand it’s difficult to collect high quality data, so I’d suggest the authors to add data from related domains where the same representation learning can be employed or augment its own data expanding the sensory system (which in principle should be cheaper).

The value of the paper could be not just to address a specific application (ballroom dance recognition) using well known feature representations and algorithmic approaches. Honestly a more interesting motivation for me would be to explore how the type of data representation employed may help overcome effects of limited information when exploring domains where only scarce data is available. Specifically in the domain of HAR, where quality datasets are very expensive to collect. For me it’d be very interesting to know which sensors, located in which part of the dances, and modelled under which representation can perform better. As I say an interesting approach could be to augment the sensory infrastructure to characterize the minimum amount of sensor data required based on different representations.

Following my last comment, the Markov chain wrapping the figure classifier does not offer a significant contribution in my opinion. Its contribution is limited to just chain predictions from models which are translating sensor signals to human movements, that last part is where (in my opinion) the contribution is. The sequence modelling could be perfectly covered by the ANN using a different network topology.

Also, when the authors claim to perform sensor classification “in real time” I expect to see some experiments covering that claim. I’m mean, there are no experiments addressing either the timing performance of the models or their power requirements. I’d recommend including in the experimentation some test of the performance of the methods, not just their accuracy.

I know it’s not possible with the current dataset, but when cross validating in HAR it’s more correct to do the splitting by user. Cross-validation based on the subjects offer a stronger vision of the generalization power of the method

To improve the reproducibility I’d recommend further info. The sensor signal was downsampled to 100 samples, but what was the original resolution? From the rotation sensor only yaw channel is used (because is “based on prior knowledge that roll and pitch are insignificant in the waltz figures included in the study), please improve this explanation. A figure explicitly addressing the input representation would be useful.

The legend of the figures should be enough to understand the figure as a whole. Please improve theIt’s honestly quite surprising that decision trees can work so well with raw features. Derived features (max, min, avg, kurtosis, skew ) tend to work better for these type of algorithms and are more commonly used.

-------------------------------------------------
Minor

Using the same terminology from sampling rate (in “2.1 Data Collection” relates to the frequency domain) and what later becomes the observation/instance (size 4x100) of a figure is a bit confusing in my opinion.

 The probability of observing the next figure is really independente of past figures sequence? Does not a dance require to include a number of figures during its execution? I’ve the feeling that the accumulated number of figures also work as a prior for the next figure.

---

### Official Review · AnonReviewer3 · 2020-10-30
**Explores some aspects of the recognition of dance figures in a sort of idealised scenario**

**Rating:** 4
**Confidence:** 5

**Review:**

This paper describes a system to classify dance figures using a wrist-worn device, exploring a number of different classifiers, and incorporating prior knowledge about dance to improve overall performance.

Weaknesses / comments / suggestions:
* Motivation / problem setting. The authors envision a scenario where the audience is informed about the individual dance moves through automated recognition. To motivate the use of ML in this setting this could be extended to illustrate the potential of ML in this area, particularly when it comes to amateurs. For example, representation learning would have the potential to allow assessment of the quality of individual moves, or the characterisation of an overall routine w.r.t. variety, where it could be used as a training aid.
* Automatic segmentation of dance moves is not explored. The authors are significantly simplifying the problem by (manually) segmenting each individual figure in the data-set, and by removing any "non-dancing" behaviour from the boundaries. I believe in a practical setting, as envisioned by the authors, this information wouldn't be readily available. Also, it would make for an interesting problem - can you recover the boundaries between the different figures if presented with a continuous signal and potentially features from an audio stream?
* Learning representations vs classification. While it is true that the NN-based approaches implicitly learn a representation of the data, they are exclusively used as classifiers in this work. It would be interesting to explore what these representations (e.g. activations in penultimate layer) actually look like, and if e.g. the dance moves that start with the same foot are similar in that representation space. "Trajectories" in that representation space would further be interesting to potentially 1) segment individual figures an 2) characterise the overall performance / variety of a complete routine.
* Related work: While this application might be novel it is nevertheless closely related to other applications of human activity recognition. The literature on this is extensive, and the paper would benefit from characterising how the problem investigated differs from settings explored in related work, e.g. for other somewhat related sports (e.g. martial arts, rock climbing).

In summary, this paper explores some aspects of the recognition of dance figures in a sort of idealised scenario. Overall the paper is clearly written, albeit for a different audience than what is typically found at ICLR. The paper describes and motivates an application, describes data collection and explores a number of (standard) methods. While it is a nice application, the suitability for this community is limited as it will be difficult to transfer findings to other application settings and there is limited technical novelty / insight.

---

### Decision · Program_Chairs · 2021-01-07
**Final Decision**

**Decision:**

Reject

**Comment:**

This paper initially received three negative reviews: 4,4,4. The main concerns of the reviewers included limited methodological novelty and an oversimplistic experimental setup. The authors did not submit their responses.
As a result, the final recommendation is reject.